# Prediction of Dynamic Behavior of Large-Scale Ground Using 1 g Shaking Table Test and Numerical Analysis

**DOI:** 10.3390/ma16186093

**Published:** 2023-09-06

**Authors:** Yong Jin, Sugeun Jeong, Daehyeon Kim

**Affiliations:** Department of Civil Engineering, Chosun University, 309 Pilmun-daero, Dong-gu, Gwangju 61452, Republic of Korea; jinyong@chosun.ac.kr (Y.J.); dkdn230@chosun.kr (S.J.)

**Keywords:** earthquake, 1 g shaking table test, numerical analysis, similarity law, scale model, dynamic behavior, response

## Abstract

Earthquake disasters can threaten human life and cause property damage. The dynamic analysis of the ground performance of the seismic field is essential. In this study, numerical analysis is used to predict the dynamic behavior and response analysis of large-scale models under different seismic waves. Firstly, the accuracy of numerical analysis is verified by a 1 g shaking table test under the same size. Then, according to the similarity law, numerical analysis is used to obtain the dynamic behavior of the model at different scales. The results show that the 1 g shaking table test results are in good agreement with the numerical analysis results and that the numerical analysis can predict the dynamic behavior of the scale model. The 1 g shaking table test provides a valuable method for evaluating the numerical analysis, which captures the complex behavior and resolves uncertainties, ultimately leading to more robust and reliable analyses.

## 1. Introduction

In recent years, the probability of global earthquakes has continued to increase. High-intensity earthquake disasters have occurred frequently, such as the magnitude 7.9 earthquake that occurred in central Nepal in April 2015, the magnitude 7.3 earthquake that occurred in Kyushu, Japan, in April 2016, the magnitude 5.4 earthquake that occurred in Pohang, South Korea, on 15 November 2017, and the magnitude 6.4 earthquake that occurred near Hualien County, Taiwan, in February 2018. Earthquakes cause the collapse of buildings and casualties, bringing endless pain and disaster to human society. The seismic wave propagated from the bedrock reaches the upper soil particles through the soil layer. The dynamic stress of the soil unit is mainly caused by shear waves propagating upward from the bedrock, and the unit will be subjected to repeated shear stresses whose magnitude and direction are constantly changing. Because of ground constraints and the uncertainty of seismic occurrence, research on sites prone to seismic action is essential.

A typical method researchers employ to simulate and analyze the dynamic behavior and response of a prototype is by performing the 1 g shaking table test as shown in Figure 1. A 1 g shaking table test can fully simulate the seismic process. It is the most direct method to study the seismic response and failure mechanism of sites in the laboratory. Most of the 1 g shaking table tests utilize prototypes and adhere to the principles of similarity, ensuring an accurate representation of real-scale ground. On the one hand, some research has focused on experimental analysis, utilizing the 1 g shaking table as the primary method for investigating and analyzing the experimental results. Kim et al. [1] aimed to evaluate the influence of soil box boundary conditions on soil dynamic behavior by comparing a rigid box (RB) and a laminar shear box (LSB) through 1 g shaking table tests. Saha et al. [2] investigated the effect of soil-structure interaction using a rigid box with permeable boundaries. Niu et al. [3] performed a 1 g shaking table test on a scaled model of the rock slope. Lin and Wang [4] conducted a shaking table test on a model slope to investigate the initiation of slope failure. The above studies all used 1 g shaking table tests to show the dynamic behaviors of the prototype, but the shortcomings of the 1 g shaking table tests were not considered, such as geostatic stress and confining stress. On the other hand, certain studies have employed numerical analysis techniques to study the 1 g shaking table models, allowing for a more detailed examination of the behavior and response of the tested systems. Kheradi et al. [5] conducted numerical analysis and 1 g shaking table tests to evaluate the effectiveness of partial ground improvement as an earthquake countermeasure for the existing box culvert described earlier. Zarnani et al. [6] developed a simplified numerical model using FLAC software to simulate the dynamic response of two reinforced earth walls constructed under 1 G conditions. Pitilakis et al. [7] conducted a significant study utilizing initial stress–strain loops to compare centrifuge test results with numerical data. Moghadam and Baziar [8] conducted a series of 1 g shaking table tests and performed FLAC 2D numerical simulations to investigate the effects of circular subway tunnels on ground motion amplification patterns. Although these studies increased the size of the model, the similarity law was not fully followed between the models, such as the input motion time.

Previous research experiences have provided valuable inspiration and suggestions for this study. Guo et al. [9] conducted a shaking table model test with a rigid box to study a symmetrical anti-bedding rock slope’s dynamic characteristics and dynamic response. The effects of dynamic parameters, seismic wave types, and weak interlayers on the slope dynamic response were considered. Aldaikh et al. [10] conducted a series of shaking table tests. The similarity law was adopted for the determination of the dynamic testing conditions of the model. In order to ensure the validity of the test results, the researchers applied the similarity law, which states that specific properties of the model should be scaled appropriately about the prototype [11,12,13]. Using the 1 g shaking table, researchers can replicate seismic motions in a controlled laboratory environment and observe how the prototype ground responds. This approach allows for a comprehensive investigation of various aspects related to seismic performance. The 1 g shaking table test has the limitation that it cannot simulate the geostatic stress and confining stress of the prototype. Zhang et al. [14] and Sadiq et al. [15] compared the dynamic analysis results with the experimental data from a centrifuge test to evaluate the dynamic behavior of a tunnel. The experimental results of the centrifuge in the free field are in good agreement with the numerical analysis results at different depths. It provides strong proof for using numerical analysis to simulate large-scale models. Therefore, it is necessary to use the simulation analysis to study the large-scale model according to the similarity law and the results of the 1 g shaking table test.

In this study, numerical analysis predicted the dynamic behavior and response of models of different scales under different seismic waves by ABAQUS and DEEPSOIL. Firstly, the 1 g shaking table test was carried out, and a model of the same size was established in the numerical analysis. The model’s dynamic behavior and response results were compared with the 1 g shaking table test and numerical analysis. A large-scale model was established through the numerical analysis, and the dynamic behavior and responses of the large-scale model using different constitutive models were obtained by combining the similarity law, as shown in Figure 2. In this way, geostatic stress and confining stress in small-scale models can be avoided to the greatest extent. The difference between this research and the previous research was that the prototype was used as the benchmark, and the accuracy of the numerical analysis was verified by comparing the 1 g shaking table experiment with the numerical analysis. The approach taken in this research is innovative and unique, as it combines a 1 g shaking table test and similarity law to provide a fresh perspective on soil behavior and the response of a full-scale model.

## 2. Materials and Methods

### 2.1. Soil Properties

The soil sample was taken from a cut slope at a construction site in Ulju-gun, Ulsan Metropolitan City. The physical properties of the soil were analyzed using specific gravity tests, grain size tests, standard Proctor tests, and relative density tests. The specific gravity of the soil was found to be 2.69, with maximum and minimum dry unit weights of 18.27 kN/m^3^ and 12.43 kN/m^3^, respectively. The optimum moisture content of the soil was determined to be 12.5%. The Atterberg limit test indicated that the Plastic Index (PI) was Non-Plastic (NP). The maximum and minimum void ratios were calculated as 1.123 and 0.443, respectively. The fine content of the soil was measured at 10.8%, and based on the Unified Soil Classification System, the soil was classified as SW-SM. For the dynamic model test, a specimen was selected from the portion of the sample passing through the No. 4 sieve after the physical property tests. The remaining sample retained in the No. 4 sieve was approximately 1%. Table 1 shows the geotechnical index properties of the specimen used in the 1 g shaking table test.

### 2.2. 1 g Shaking Table Test

#### 2.2.1. Experimental Equipment

This study utilizes several experimental components to investigate the dynamic behavior of soil. Figure 3 illustrates the schematic diagram of the 1 g shaking table test system, which was the core component for conducting the experiments and obtaining insights into the soil’s dynamic properties. The major equipment includes a 1 g shaking table test system, a laminar shear box, a data logger, and accelerometers.

The uniaxial shaking table tests the structure’s response and verifies the seismic simulation. The equipment simulates a variety of ground motions, including the reproduction of recorded seismic timelines and solid liquefaction and vibration tests. An electronic servo control valve with a high-precision closed-loop PID controller powers the system. The specifications of the shaking table are shown in Table 2.

The laminar shear box consists of 12 aluminum frames that move independently horizontally, as shown in Figure 4a. Each frame can simulate horizontal shear motion’s infinitely expanding ground boundary conditions. The dimensions of the structure are 2000 mm (W) × 600 mm (L) × 600 mm (H), each frame is 4.5 cm thick, and the spacing between the frames is about 0.5 cm. The natural period of the empty box is 0.04–0.05 s.

Data logging is the measurement and recording of physical or electrical parameters over a period. Accelerometers are used to capture seismic acceleration. An ARF-20A acceleration transducer was used as the accelerometer for this study, which can be measured up to 20 m/s^2^. The data logger used the 24-channel SDL-350R model and was compatible with ARF-20A. The data storage interval is up to 0.005 s. Figure 4b,c show the data logger and accelerometer used in this study.

#### 2.2.2. Experimental Method

The artificial seismic waves, Hachinohe seismic waves, and Ofunato seismic waves, which correspond to the peak ground acceleration of 0.2 g, were used to determine the fundamental behavior of the ground. The artificial seismic wave is the synthetic seismic wave combining the Gyeongju—Pohang earthquake, using the empirical Green’s function based on the raw data measured at the Kori Nuclear Power Plant. As the artificial seismic wave has different periodic components, high amplification occurs. This is why the artificial seismic wave was selected to evaluate the effects of boundary conditions in the case of high amplification. The Hachinohe wave is a long-period wave and the Ofunato wave is a short-period wave. The seismic waves in this study are shown in Figure 5.

The experiments were divided into three sets, focusing on slope models. For each set of experiments, three different input motions were used. To obtain the dynamic behavior of the models in this study, 1 g shaking table tests were used. Accurate experimental results can be obtained by carefully conducting each experimental step. First, soil samples were put into the soil box and compacted into a five-centimeter-thick layer per layer in order to distribute the soil as evenly as possible throughout the entire box. Then, accelerometers were installed in the middle of each soil layer. Finally, the soil of the flat ground was gently excavated to form a slope. It was carried out to ensure the stability of the sloping ground. In order to carry out the 1 g shaking table test, seismic waves were applied to the soil model. Accelerometers were used to measure the accelerations experienced by the model during the test, and the data logger was utilized to record the experimental results. Figure 6 shows the location of the accelerometers set in the slope model in the 1 g shaking table test.

### 2.3. Numerical Analysis

This study used different scale models to study the ground’s dynamic behavior and response analysis. The 1 g shaking table test’s experimental results were studied using the similarity law. The following parameters were selected as references for this research in Table 3. Under the similarity law, the positions of the large-scale models were selected with the same proportion, and the input seismic waves were also converted. For example, the depth of the 20 × 6 m (λ = 10) model is 0.5 m. The times of the artificial seismic wave, Hachinohe seismic wave, and Ofunato seismic wave are 151.42, 88.8, and 84.35, respectively.

#### 2.3.1. DEEPSOIL Program

DEEPSOIL is a 1-D site response analysis program with a graphical user interface that can perform 1-D nonlinear analysis. The depth of each model is 2 m (1 time), 4 m (2 times), 10 m (5 times), 20 m (10 times), 50 m (25 times), and 100 m (50 times). In total, 18 models were established in DEEPSOIL. The artificial seismic wave, Hachinohe seismic wave, and Ofunato seismic wave were applied. It should be noted that only flat part analysis results were used in the study because DEEPSOIL does not work well on slopes. In DEEPSOIL models, each model was divided into six layers. The soil parameters in DEEPSOIL are unit weight, effective vertical stress, shear wave velocity, and shear strength. Effective vertical stress increases linearly with depth and is only related to unit weight and depth and the parameters are shown in Table 4.

The constitutive model used by DEEPSOIL is the Darendeli model. Darendeli [17] proposed the model as follows:(1)$$\frac{G}{G_{max}}=\frac{1}{1+{\left(\frac{\gamma }{\gamma_{r}}\right)}^{a}}$$
where, G= shear modulus; $G_{max}=$ small-strain shear modulus, γ= shear strain, $\gamma_{r}=$ reference strain, a= curvature coefficient.

Regarding the calculation of shear strength, guidance has been given in the DEEPSOIL user manual. The GO/H model necessitates inputting the soil layer’s shear strength to represent the soil’s large-strain behavior. Frequency changes according to the change of each layer. The calculation equation is:(2)$$f=V_{s}/4H,$$
where, f= frequency of the layer (Hz), $V_{s}=$ shear wave velocity (m/s), H= the depth of the layer (m).

#### 2.3.2. Finite Element Analysis

ABAQUS CAE is a finite element analysis and multi-physics engineering simulation software. In this study, 36 models were established. The slope models were established in ABAQUS using the Mohr–Coulomb and Borja models. The size models modeled by ABAQUS here are 2 × 0.6 m (laboratory size model), 4 × 1.2 m (λ = 2), 10 × 3 m (λ = 5), 20 × 6 m (λ = 10), 40 × 12 m (λ = 20), 100 × 30 m (λ = 50). Artificial seismic waves, Hachinohe seismic waves, and Ofunato seismic wave incident waves are applied at the bottom of each model, respectively.

The constitutive models used are the Mohr–Coulomb and Borja models. The Mohr–Coulomb model and Borja model (User Material) are constitutive models commonly used to describe the mechanical behavior of materials. They differ in their approach to modeling shear strain and the resulting behavior of materials. The Mohr–Coulomb model is more straightforward and assumes a linear relationship between shear stress and strain. In contrast, the Borja model is a more advanced model that incorporates nonlinear relationships and can simulate a broader range of soil behavior, as shown in Figure 7. The parameters used in the Mohr–Coulomb model are cohesion yield stress, internal friction angle, and dilatancy angle, as shown in Table 5.

The Umat model used in ABAQUS is the Borja model. This model was proposed initially by Borja et al. [18].
(3)$$\frac{G}{G_{max}}=1-\frac{3}{2\gamma_{0}}\int_{0}^{2\tau_{0}}{\left[h{\left(\frac{\frac{R}{\sqrt{2}}+\tau_{0}-\tau }{\tau }\right)}^{m}+H_{0}\right]}^{-1}d\tau$$
where, G= shear modulus; $G_{max}=$ small-strain shear modulus, $\tau_{0}=$ initial shear stress, $\gamma_{0}=$ reference strain, $h,m,H_{0}=$ level of hardening, R= the radius of the bounding surface.

Table 6 presents the soil parameters utilized in the ABAQUS software. These parameters reflect the calibrated values obtained through the calibration process, ensuring that the Borja model accurately represents the soil behavior for the specified conditions in the ABAQUS simulations. Based on the test result of soil, the parameters for the bounding surface plasticity model were chosen as R = 0.04 G_max_, h = 0.16 G_max_, m = 0.8.

The model was divided into a mesh with dimensions of 0.05 × 0.05 m. The infinite boundary part employs CINPE4 elements. It is a 2D model with only horizontal and vertical boundaries. The bottom boundary only has horizontal acceleration and cannot carry motion in the vertical direction. In order to eliminate the impact of the boundary on model analysis results, the two sides of the boundary were redesigned into infinite boundaries, and the boundary conditions were consistent with the laminar shear box as much as possible, as shown in Figure 8.

## 3. Results and Discussion

This section used DEEPSOIL and ABAQUS programs to analyze and compare different seismic input motions. In the 1 g shaking table test, the top of the model is often where the acceleration was most amplified. In this study, the accelerations of the flat and slope ground surfaces were consistent with the locations of accelerometer 5 and accelerometer 8 in Figure 6.

The 1 g shaking table test results and numerical analysis results were compared through root mean square error. The RMSE values are calculated to assess the overall accuracy and reliability of the numerical analysis. When the analysis value of RMSE is close to 0, it is more consistent with the experimental value. In seismology or geotechnical engineering, RMSE reference values for simulated acceleration-time histories range from 0 to 0.3 [19,20]. Artificial seismic waves contain the characteristics of long-period and short-period waves. In the following, the results are presented only with artificial seismic waves.

### 3.1. Acceleration-Time History

The difference between the experimental and numerical analysis results can be more clearly understood by comparing a small acceleration segment. Figure 9 and Figure 10 show the part of the acceleration-time history of the 1 g shaking table test and numerical analysis with the artificial seismic wave in different scales.

The analysis of the acceleration variation between the experimental results and numerical results was conducted to compare the predicted ground acceleration values and accelerations amplified with depth. At the same depth, the acceleration of the sloping part is greater than that of the flat part. Experimental and numerical data cannot be directly compared because they are very close. Therefore, values were analyzed using the RMSE method. The differences in the numerical analysis were confirmed based on the experimental results in this study. The numerical analysis results are in good agreement with the experimental results. In the flat ground, the RMSE analysis values of DEEPSOIL, ABAQUS (M-C), ABAQUS (Borja) results, and experimental results are 0.0184, 0.0527, and 0.0460, respectively. In the sloping ground, the RMSE analysis values of ABAQUS (M-C), ABAQUS (Borja) results, and experimental results are 0.1017 and 0.0512, respectively. It shows that the numerical analysis results are very close to the experimental results, which proves that the numerical analysis method is feasible. At the same time, the accuracy of the 1 g shaking table model was also verified.

### 3.2. Peak Ground Acceleration

Peak ground acceleration (PGA) is a measure used to quantify the maximum acceleration experienced by the ground during an earthquake. It represents the highest value of ground acceleration recorded at a specific location during the seismic event. Figure 11 is the PGA of the experiment result and numerical analysis result of the flat part with the artificial seismic wave in different scales.

The relationship between PGA and depth highlighted the importance of considering site-specific soil properties and layering effects in seismic hazard assessments. Experiment and numerical analysis results showed a consistent tendency to increase as the depth became shallower. Whether it is the experimental results or the numerical analysis results, with the increase in depth, the PGA presents an increasing trend. By examining the differences in PGA values between numerical analyses, insights can be gained into the accuracy and reliability of these models in estimating ground shaking. The findings suggest that DEEPSOIL may be more suitable for capturing the ground-shaking characteristics and providing reliable estimates of PGA compared to ABAQUS in the context of this study.

### 3.3. Spectral Acceleration

Spectral acceleration is a measure used in earthquake engineering to quantify the amplitude of ground motion at different frequencies during an earthquake. It represents the maximum acceleration that a specific structure or location experiences at a particular frequency. Figure 12 and Figure 13 show the spectral acceleration of the experiment and analysis with the artificial seismic wave at different scales.

By comparing the spectral acceleration, when it is less than the natural period, for a flat part and a sloping part, the amplification factor gradually decreases with the increase in the natural period for the flat part. The ABAQUS (Mohr–Coulomb model) analysis value is the highest, followed by the ABAQUS (Borja model), DEEPSOIL, and experimental values. The ABAQUS value is more significant than that of the experimental analysis at the sloping part. One of the reasons for the difference between the ABAQUS and DEPSOIL analysis may be due to the difference in constitutive models. The Mohr–Coulomb model was used in ABAQUS, and the Darendeli nonlinear model was used in DEEPSOIL. However, the ABAQUS (Borja model) result is very close to DEEPSOIL. In general, the experimental results are very close to the numerical results.

Peak spectral acceleration can be predicted based on the relationship between frequency and size. The accuracy of modeling using ABAQUS is verified by calculating the period and frequency of the model. Therefore, the values in the numerical model conform to the equations and scale factor.

### 3.4. The Stress–Strain Curve of Large-Scale Models

In this study, the surface layers of the model will be selected to obtain the shear stress–strain value from the numerical analysis. Figure 14 and Figure 15 show the stress–strain curve of the Mohr–Coulomb model, Darendeli model, and Borja model of the 1-time and 50-times model, respectively.

In the one-time model, when the strain is less than 0.03%, the stress–strain curve of the Mohr–Coulomb model is linear, which indicates that the model has only elastic ranges. The Darendeli model and Borja model can also capture stress and strain well. Numerically, the stress and strain of the Darendeli model were smaller than the other two, which was based on the difference in the parameters of the constitutive model.

Moreover, the hysteretic curves of the Darendeli model and the Borja model also appear to be unable to follow the Masing criterion because as the strain increases, the damping of the model increases rapidly, and energy is absorbed and attenuated. In the 50-times model, when the strain is more significant than 0.7%, the Darendeli model cannot capture the shear strain well, which can be considered as the Darendeli model reaching the limit of obtaining strain in this study. However, the Mohr–Coulomb model and Borja model can obtain shear strain. It is essential to understand the dynamic behavior of the ground.

## 4. Conclusions

This study has investigated the feasibility of the 1 g shaking table test for predicting the dynamic behavior of the full-scale ground. According to the 1 g shaking table test results, the similarity law was used to analyze the large-scale model in numerical analysis. The conclusions of this study are as follows:By comparing RMSE results, the experimental results were in good agreement with the numerical analysis results in terms of consistency. The dynamic behavior of the slope model from the numerical analysis was consistent with that from the 1 g shaking table test. It was shown that the laminar shear box can minimize the influence of boundaries on the dynamic behavior of soil. The laminar shear box was evaluated to perform well for the slope model. The results of the ABAQUS analysis were in good agreement with those of the experimental analysis for the slope model.For different constitutive models, the numerical analysis results were still slightly different. For the flat ground, DEEPSOIL results were closer to the experimental results. For the slope model, the Borja model gave better results than the Mohr–Coulomb model. The input parameters of different constitutive models are different, which is why different numerical analysis results exist.Numerical analysis was conducted to obtain stress–strain curves for different constitutive models. The numerical analysis results indicated that the Daredneli model did not accurately capture the behavior under high-strain conditions in the dynamic analysis. On the other hand, the Mohr–Coulomb and Borja models performed better in representing the stress–strain response. It highlights the advantage of using nonlinear and elastoplastic models in their respective applicable regions. The Darendeli model sometimes needs to adequately capture the dynamic behavior of soils under more significant strains, but the Borja model does not.The 1 g shaking table test provides a valuable method to evaluate numerical analysis, capture complex behavior, and resolve uncertainties, ultimately leading to more robust and reliable analysis and enhancing the value of the 1 g shaking table test.In this study, extensive numerical analysis has been performed to overcome the size limitation of the 1 g shaking table test in predicting the dynamic behavior of real-scale ground. Combining the results of numerical analysis and the 1 g shaking table test, as well as a series of theories, such as the similarity law, the 1 g shaking table experiment can replace the centrifuge test. An equation has been developed to obtain the natural frequency of the real-scale ground. In actual earthquake engineering, the natural frequency can be obtained by this method. The prediction and analysis of the dynamic behavior of large-scale ground by numerical analyses along with the 1 g shaking table test is significant.

## Figures and Tables

**Figure 1 materials-16-06093-f001:**
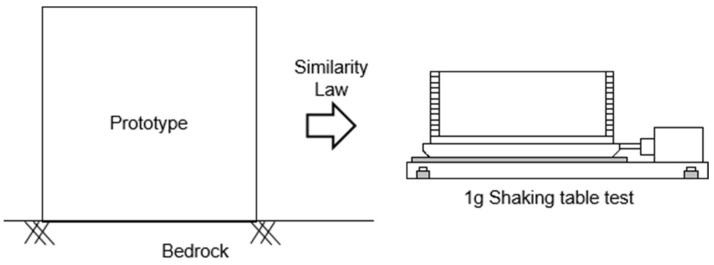
Realization of prototype dynamic behavior and response in 1 g shaking table test.

**Figure 2 materials-16-06093-f002:**
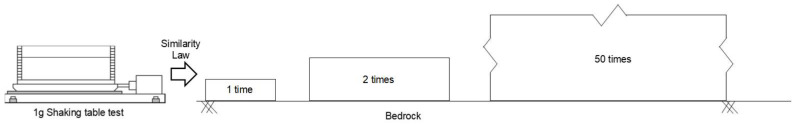
Simulation of the dynamic behavior and responses of large-scale models through 1 g shaking table test.

**Figure 3 materials-16-06093-f003:**
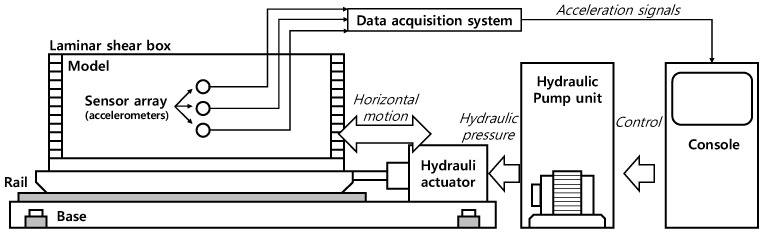
The experimental system used in this study [16].

**Figure 4 materials-16-06093-f004:**
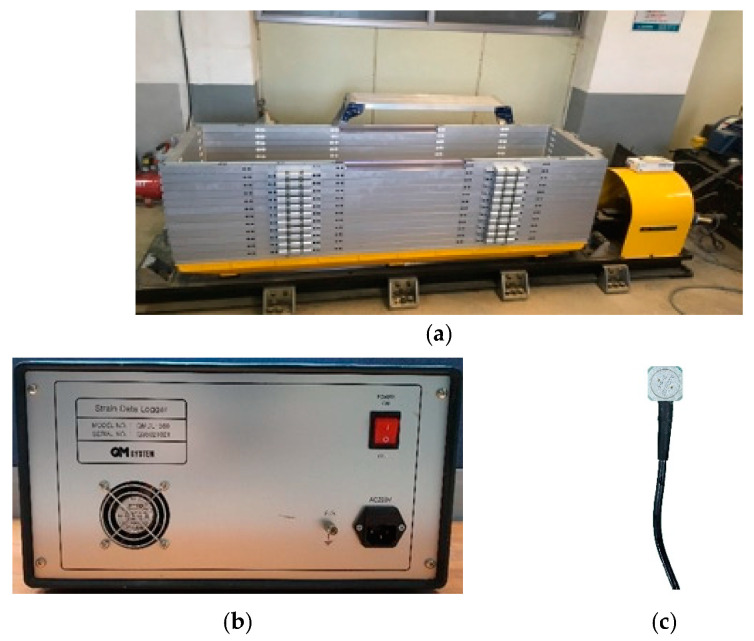
Experimental equipment: (**a**) Laminar shear box; (**b**) Data logger; (**c**) Accelerometer.

**Figure 5 materials-16-06093-f005:**
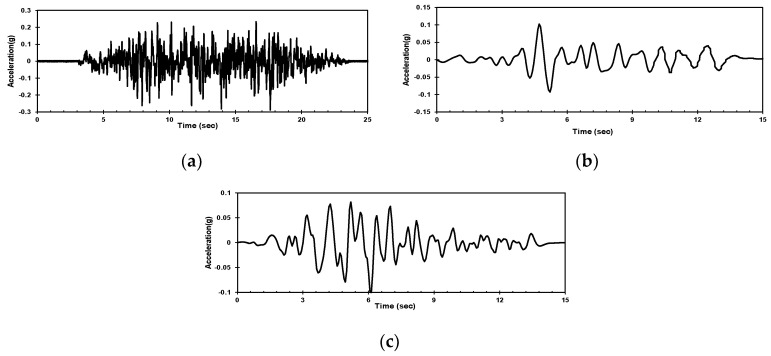
Acceleration-time history of the input ground motions used in this study: (**a**) Artificial seismic wave; (**b**) Hachinohe seismic wave; (**c**) Ofunato seismic wave.

**Figure 6 materials-16-06093-f006:**
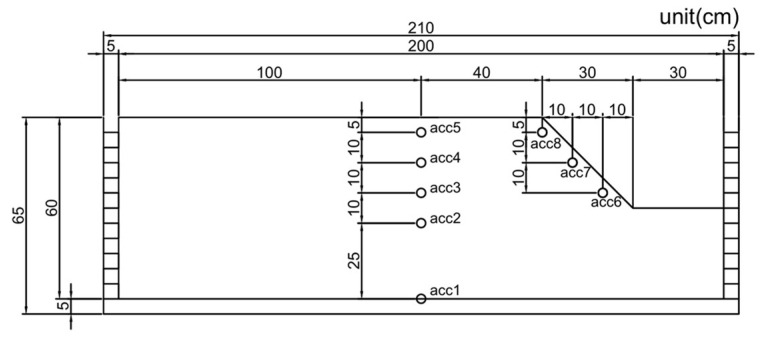
Accelerometer setting in the slope model.

**Figure 7 materials-16-06093-f007:**
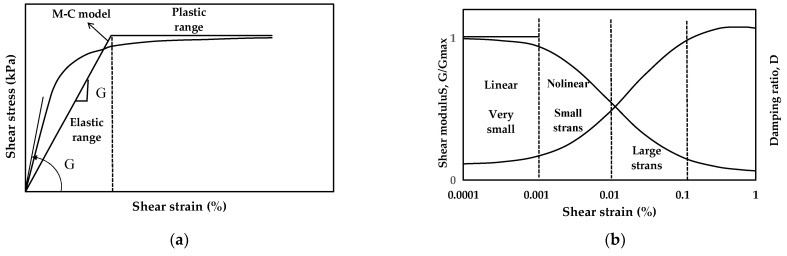
Acceleration-time history of the input ground motions used in this study: (**a**) A typical shear stress—shear strain relationship of soil; (**b**) Normalized shear modulus and damping ratio.

**Figure 8 materials-16-06093-f008:**
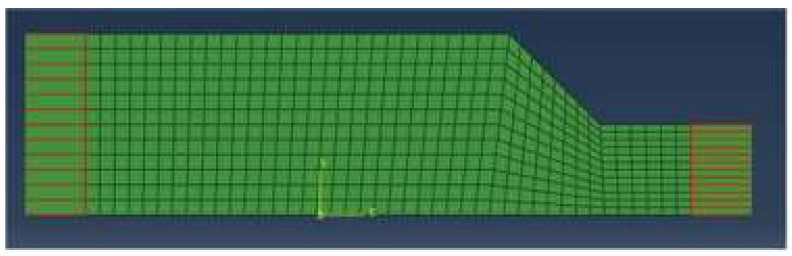
Infinite boundary model.

**Figure 9 materials-16-06093-f009:**
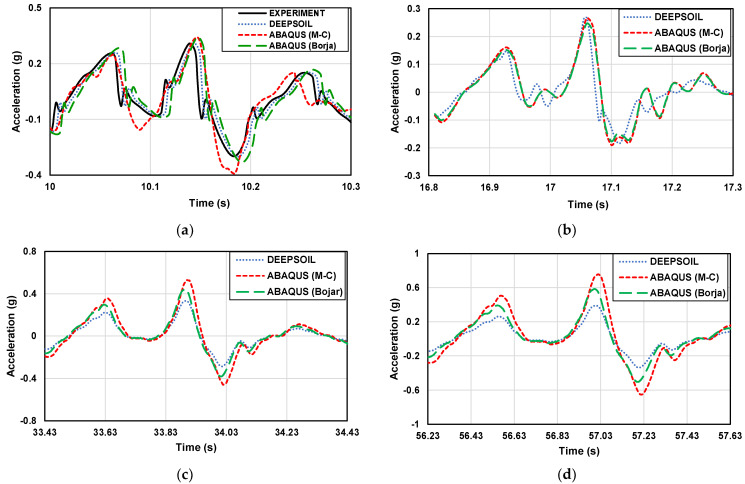
Part of the acceleration-time history of the artificial seismic wave in flat ground. (**a**) 2 × 0.6 m (1 time); (**b**) 4 m × 1.2 m (2 times); (**c**) 10 m × 3 m (5 times); (**d**) 20 m × 6 m (10 times); (**e**) 50 × 15 m (25 times); (**f**) 100 × 30 m (50 times).

**Figure 10 materials-16-06093-f010:**
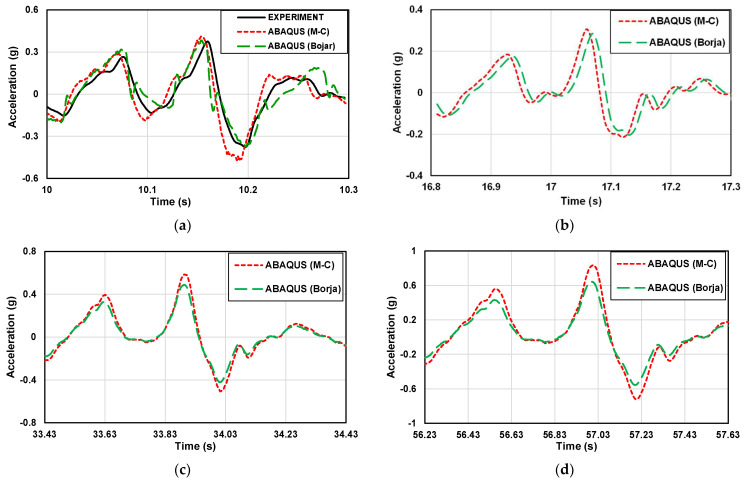
Part of the acceleration-time history graph of the artificial seismic wave in the sloping ground. (**a**) 2 × 0.6 m (1 time); (**b**) 4 m × 1.2 m (2 times); (**c**) 10 m × 3 m (5 times); (**d**) 20 m × 6 m (10 times); (**e**) 50 × 15 m (25 times); (**f**) 100 × 30 m (50 times).

**Figure 11 materials-16-06093-f011:**
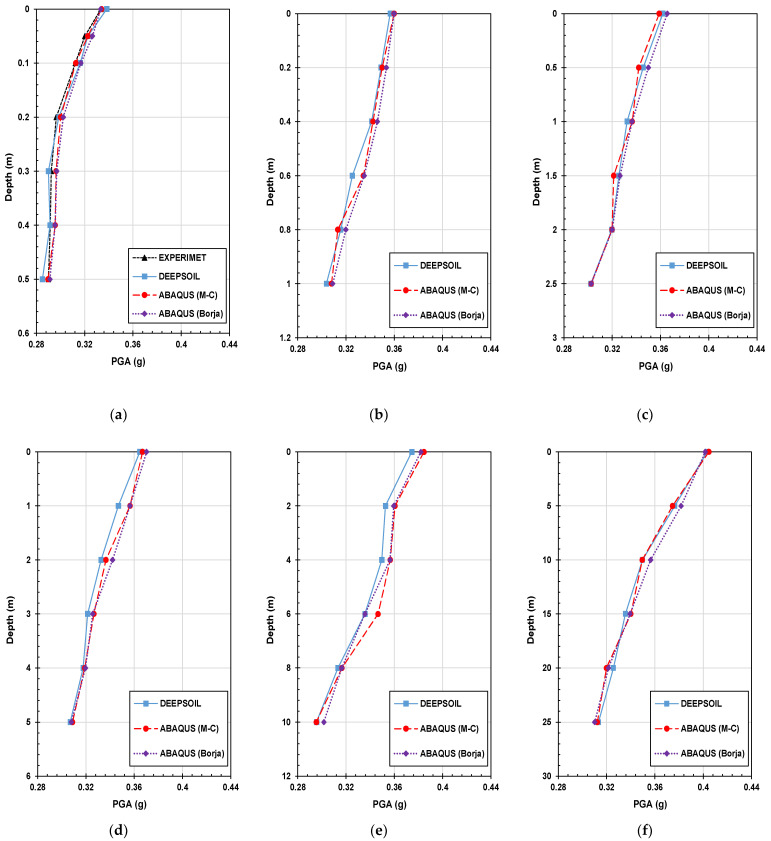
PGA profile for experiment and numerical analysis with the artificial seismic wave. (**a**) 2 × 0.6 m (1 time); (**b**) 4 m × 1.2 m (2 times); (**c**) 10 m × 3 m (5 times); (**d**) 20 m × 6 m (10 times); (**e**) 50 × 15 m (25 times); (**f**) 100 × 30 m (50 times).

**Figure 12 materials-16-06093-f012:**
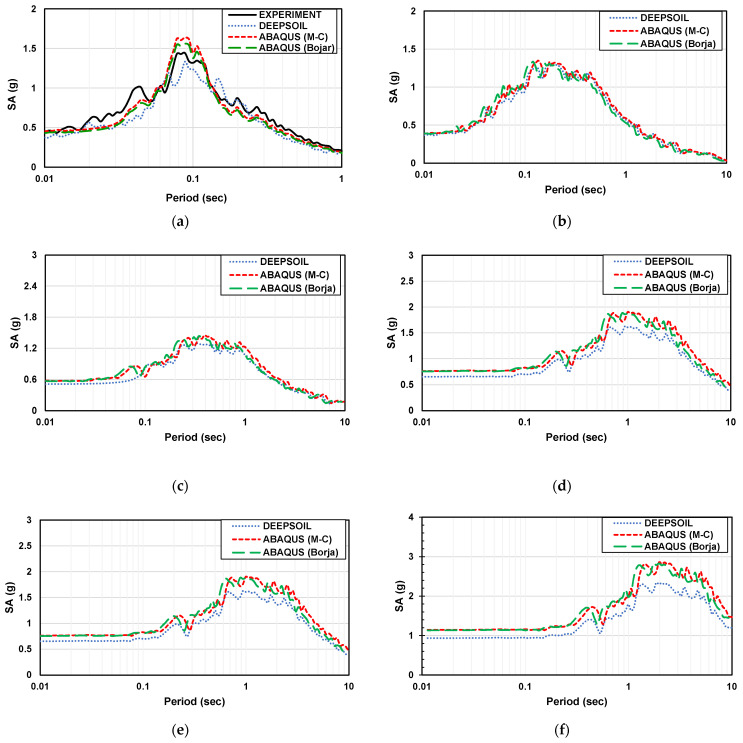
The spectral acceleration of Artificial seismic wave in the flat ground. (**a**) 2 × 0.6 m (1 time); (**b**) 4 m × 1.2 m (2 times); (**c**) 10 m × 3 m (5 times); (**d**) 20 m × 6 m (10 times); (**e**) 50 × 15 m (25 times); (**f**) 100 × 30 m (50 times).

**Figure 13 materials-16-06093-f013:**
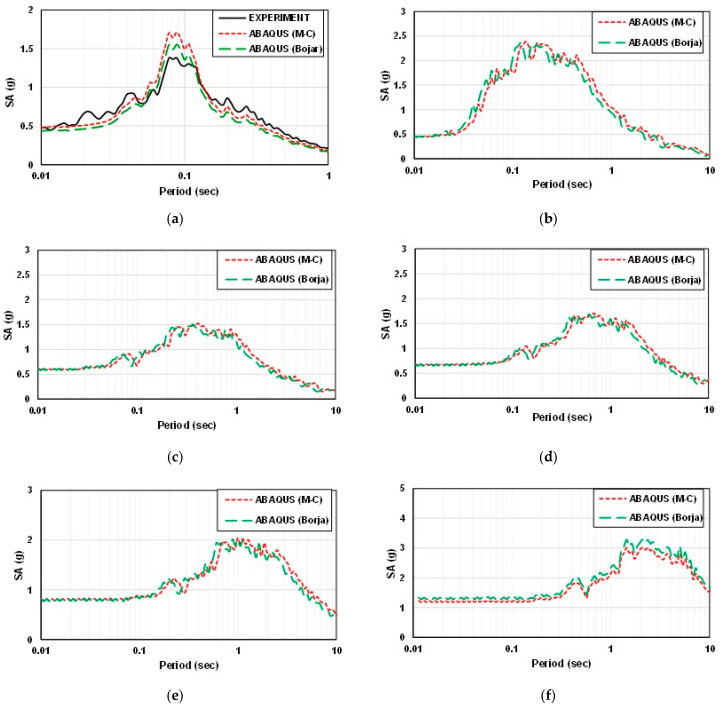
The spectral acceleration of the artificial seismic wave in the sloping ground (**a**) 2 × 0.6 m (1 time); (**b**) 4 m × 1.2 m (2 times); (**c**) 10 m × 3 m (5 times); (**d**) 20 m × 6 m (10 times); (**e**) 50 × 15 m (25 times); (**f**) 100 × 30 m (50 times).

**Figure 14 materials-16-06093-f014:**
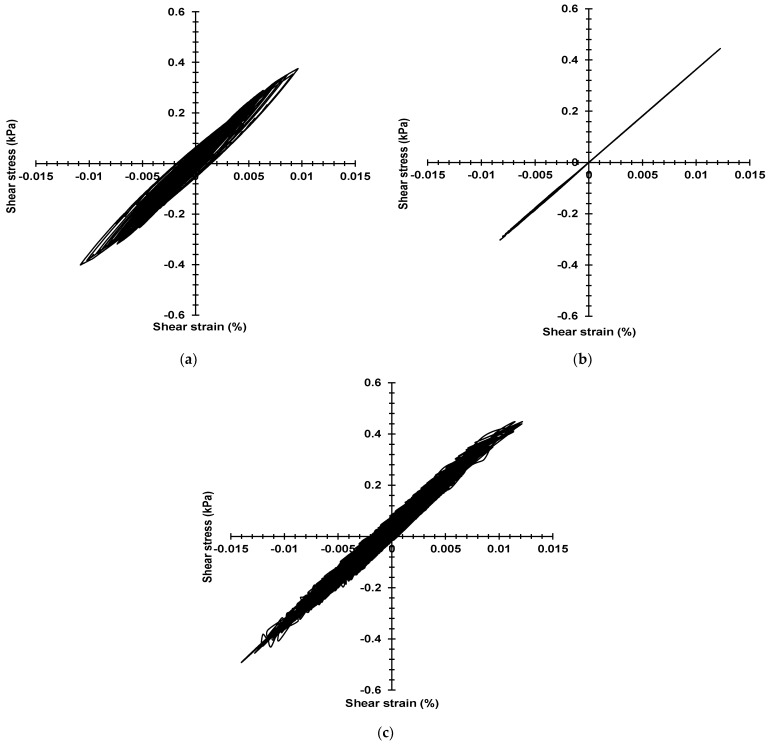
The stress–strain curve of 2 × 0.6 m (1-time) model: (**a**) Darendeli model; (**b**) Mohr–Coulomb model; (**c**) Bojar model.

**Figure 15 materials-16-06093-f015:**
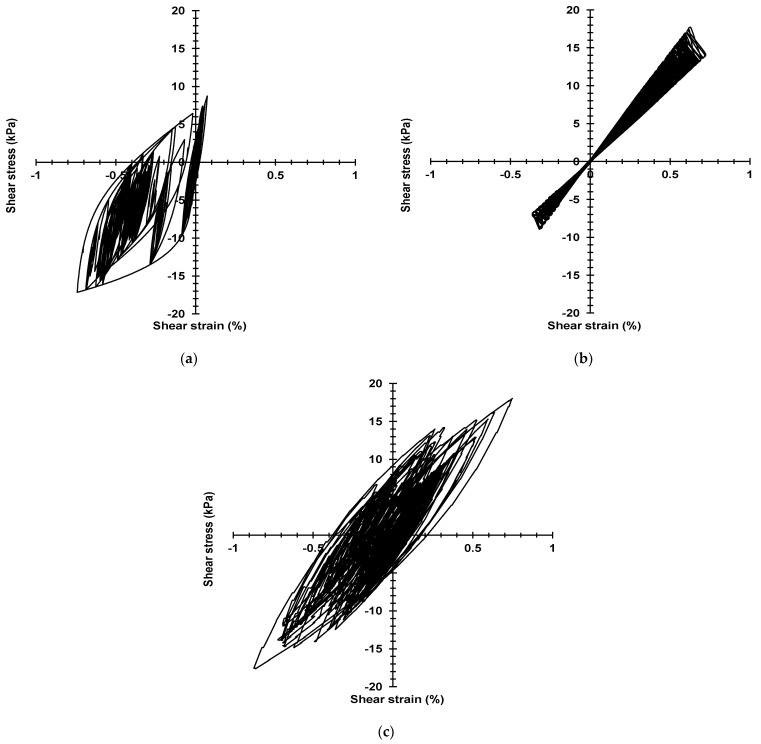
The stress–strain curve of 100 × 30 m (50 times) model: (**a**) Darendeli model; (**b**) Mohr–Coulomb model; (**c**) Bojar model.

**Table 1 materials-16-06093-t001:** Geotechnical index properties of the soil used in this study.

Parameter	Value	Parameter	Value
No. 200 Passing (%)	10.8	emax	1.123
Gs	2.69	emin	0.443
OMC (%)	12.5	rd max (kN/m^3^)	18.27
PI (%)	NP	rd min (kN/m^3^)	12.43
USCS	SW-SM		

**Table 2 materials-16-06093-t002:** Uniaxial shaking table parameters and operating specifications.

Item	Specification
Table size (mm)	2000 × 600
Maximum acceleration (g)	1
Full play load (kg)	1800
Payload capacity (kg)	5000
Operating frequency (Hz)	10

**Table 3 materials-16-06093-t003:** Similarity law was used in this study.

Item	Specification	Item	Specification	Item	Specification
Mass density	1	Length	λ	Acceleration	1
Frequency	λ^−1^	Shear wave velocity	λ^−0.5^	Stress	λ
Modulus	1	Time	λ^0.75^	Strain	1

λ is the similarity ratio of the large-scale model to the laboratory-size model.

**Table 4 materials-16-06093-t004:** Soil input parameters used in DEEPSOIL (Darendeli model).

Parameter	Value
Unit weight (kN/m^3^)	17.658
OCR	1
N	10
K_0_	0.5
Frequency	$f=V_{s}/4H$,

**Table 5 materials-16-06093-t005:** Soil input parameters used in ABAQUS (M-C).

Parameter	Value
Density (kg/m^3^)	1800
Poisson’s ratio	0.3
Poisson’s ratio	0.3
Internal friction angle (°)	27.7
Cohesion yield stress (kN)	10
Dilatancy angle (°)	24.4

**Table 6 materials-16-06093-t006:** Soil input parameters used in ABAQUS (Borja).

Parameter	Value	Parameter	Value
Density (kg/m^3^)	1800	Young’s modulus (Pa)	$E=2\rho V^{2}(1+\upsilon )$
Poisson’s ratio	0.3	h	2 MPa
m	0.8	R	50 kpa
Omega	0.414	xi	0.0785
H_0_	0		

## Data Availability

The data are not publicly available but can be made available upon request.

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
