# Peer review of "Prediction of Dynamic Behavior of Large-Scale Ground Using 1 g Shaking Table Test and Numerical Analysis"

_materials, 2023, doi:10.3390/ma16186093_

Round 1

Reviewer 1 Report

1. The paper is addressed to the prediction of the dynamic behaviour of large-scale ground. Since earthquake-related disasters can threaten human life, a comprehensive study of the behaviour of structures and grounds under such phenomena, as well as the justification of numerical models and simulation methods, is an urgent area of research. 

2. The paper contains the following test and modelling results obtained by the authors:

- Acceleration-time history;

- Peak ground acceleration;

- Spectral acceleration;

- The stress-strain curve of large-scale models.

The modelling results generally capture the experimental curves. However, the analysis of the results as well as the justification of the transition from scale models to prototype needs to be described in more detail.

3. Comments on the article:

- The order of citation of references is broken in the article. For example, [1-7] and then immediately [14]. Also, such a citation style when seven articles are cited in one row is not acceptable. In fact, the research results of these seven articles are not analysed.

- The quality of figures should be improved. For example, see figures 3, 6, 7.

- Figure 3 refers to the authors' published article. Therefore, provide in the Introduction a more detailed justification of the novelty of the research in the article submitted for review. 

- The conclusions are not specific and do not reflect the results obtained by the authors. The article should be improved in terms of the discussion of the results.

- Although the sources in the references are applicable to the problem investigated by the authors, the results presented in these sources are not analysed in detail.

Summary: the article contains interesting results of the experimental study and numerical modelling, but it requires highlighting of new findings of the authors.

Author Response

Dear Editors and Reviewers,

Thank you very much for your help in processing the reviews of our manuscript (Manuscript ID materials-2566226). We have carefully read the professional comments from you and reviewers, and found that these suggestions are helpful for us to improve our manuscript. On the basis of the enlightening questions and helpful advices, we have now completed the revisions of our manuscript. The itemized responses to the reviewers’ comments are listed in the succeeding sheets. We hope that all these corrections and revisions would be satisfactory. Thanks a lot, again.

Title: Prediction of Dynamic Behavior of Large-Scale Ground Using 1 g shaking Table Test and Numerical Analysis

Comments and Suggestions for Authors

  1. The paper is addressed to the prediction of the dynamic behaviour of large-scale ground. Since earthquake-related disasters can threaten human life, a comprehensive study of the behaviour of structures and grounds under such phenomena, as well as the justification of numerical models and simulation methods, is an urgent area of research. 
  2. The paper contains the following test and modelling results obtained by the authors:

- Acceleration-time history;

- Peak ground acceleration;

- Spectral acceleration;

- The stress-strain curve of large-scale models.

The modelling results generally capture the experimental curves. However, the analysis of the results as well as the justification of the transition from scale models to prototype needs to be described in more detail.

  1. Comments on the article:

- The order of citation of references is broken in the article. For example, [1-7] and then immediately [14]. Also, such a citation style when seven articles are cited in one row is not acceptable. In fact, the research results of these seven articles are not analysed.

Response: Thank you for your guidance, the literature citations have been revised.

- The quality of figures should be improved. For example, see figures 3, 6, 7.

Response: Modifications have been made to the figures.

- Figure 3 refers to the authors' published article. Therefore, provide in the Introduction a more detailed justification of the novelty of the research in the article submitted for review. 

Response: The differences between this study and previous studies have been pointed out in the introduction.

- The conclusions are not specific and do not reflect the results obtained by the authors. The article should be improved in terms of the discussion of the results.

Response: The discussion and analysis have been added to the conclusion. Thanks for your comments.

- Although the sources in the references are applicable to the problem investigated by the authors, the results presented in these sources are not analysed in detail.

Response: The results of some specific analyzes have been added, 

Summary: the article contains interesting results of the experimental study and numerical modelling, but it requires highlighting of new findings of the authors.

Reviewer 2 Report

The following are the comments to incorporate for the betterment of the manuscript.

1.      References are not well distributed in literature review. They are focused at one place, for example, ref [1-7] and [8-14].

2.      The literature review can be modified and well explained. It is very hard to understand Figure1 and 2. Represent the names on the figures.  

3.      Mention how or what tool is used for numerical analysis? Last paragraph of section 1

4.      Figure 6 is not clear

5.      Do the chosen numerical models consider strain rate effect? As this model is plastic model and the loading is dynamic? Explain

6.      Many abbreviations are not mentioned for examples RMSE?

7.      Fig 10-15, technical discussions is not given importance. Discuss the figures scientifically and technically

It is reasonable

Author Response

Dear Editors and Reviewers,

Thank you very much for your help in processing the reviews of our manuscript (Manuscript ID materials-2566226). We have carefully read the professional comments from you and reviewers, and found that these suggestions are helpful for us to improve our manuscript. On the basis of the enlightening questions and helpful advices, we have now completed the revisions of our manuscript. The itemized responses to the reviewers’ comments are listed in the succeeding sheets. We hope that all these corrections and revisions would be satisfactory. Thanks a lot, again.

Title: Prediction of Dynamic Behavior of Large-Scale Ground Using 1 g shaking Table Test and Numerical Analysis

Comments and Suggestions for Authors

The following are the comments to incorporate for the betterment of the manuscript.

  1. References are not well distributed in literature review. They are focused at one place, for example, ref [1-7] and [8-14].

Response: The references have been revised, thanks for your guidance.

  1. The literature review can be modified and well explained. It is very hard to understand Figure1 and 2. Represent the names on the figures.  

Response: Sorry for not being clear, Figure1 and 2 have been renamed.

  1. Mention how or what tool is used for numerical analysis? Last paragraph of section 1

Response: The tools used have been added, using numerical analysis ABAQUS and DEEPSOIL.

  1. Figure 6 is not clear

Response: Figure 6 has been modified.

  1. Do the chosen numerical models consider strain rate effect? As this model is plastic model and the loading is dynamic? Explain

Response: Darendeli model, mohr-coulomb model and Borja model do not explicitly consider strain rate effects. Mohr-Coulomb model is an elastic-plastic model. When a dynamic loading is applied to the model, the shear strain will continue to increase as the shear stress increases. When the shear strain reaches the plastic limit, the model changes from elastic to plastic. In this study, the model did not enter the plastic region, so the efficiency of change was not considered. For the Darendeli model and Borja model, the shear strain increases nonlinearly as the shear stress increases.

  1. Many abbreviations are not mentioned for examples RMSE?

Response: All abbreviations have been checked and all abbreviations explained.

  1. Fig 10-15, technical discussions is not given importance. Discuss the figures scientifically and technically

Response: The discussion and analysis have been added to the conclusion. Thanks for your comments.

Round 2

Reviewer 2 Report

Accept in its present form